# B Cell-Derived and Non-B Cell-Derived Free Light Chains: From Generation to Biological and Pathophysiological Roles

**DOI:** 10.3390/ijms26157607

**Published:** 2025-08-06

**Authors:** Linyang Li, Huining Gu, Xiaoyan Qiu, Jing Huang

**Affiliations:** 1Department of Immunology, School of Basic Medical Sciences, and NHC Key Laboratory of Medical Immunology, Peking University, Beijing 100191, China; 2210301238@stu.pku.edu.cn (L.L.); guhuining98@bjmu.edu.cn (H.G.); qiuxy@bjmu.edu.cn (X.Q.); 2PUHSC Primary Immunodeficiency Research Center, School of Basic Medical Sciences, Peking University, Beijing 100191, China

**Keywords:** immunoglobulin, free light chain, non-B cell, Igκ, Igλ

## Abstract

Immunoglobulin light chains are essential components of intact immunoglobulins, traditionally believed to be produced exclusively by B cells. Physiologically, excess light chains not assembled into intact antibodies exist as free light chains (FLCs). Increasingly recognized as important biomarkers for diseases such as multiple myeloma, systemic amyloidosis, and light chain-related renal injuries, FLCs have also been shown in recent decades to originate from non-B cell sources, including epithelial and carcinoma cells. This review primarily focuses on novel non-B cell-derived FLCs, which challenge the conventional paradigms. It systematically compares B cell-derived and non-B cell-derived FLCs, analyzing differences in genetic features, physicochemical properties, and functional roles in both health and disease. By elucidating the distinctions and similarities in their nature as immune regulators and disease mediators, we highlight the significant clinical potential of FLCs, particularly non-B cell-derived FLCs, for novel diagnostic and therapeutic strategies.

## 1. Introduction

Free light chains (FLCs), a slight excess of kappa (κ) and lambda (λ) light chains (LCs) without binding to heavy chains (HCs) and therefore existing outside the tetrameric structure of intact immunoglobulins, have emerged as critical players in both physiological and pathological processes, though once regarded as mere byproducts of immunoglobulin synthesis. Traditionally, immunoglobulins, composed of heavy and light chains, were thought to be exclusively produced by B lymphocytes, serving as central mediators of humoral immunity. However, mounting evidence over the past two decades has challenged this paradigm, revealing that non-B cell lineages, including epithelial cells, neurons, and cardiomyocytes, also synthesize and secrete immunoglobulins and FLCs [1]. These discoveries not only redefine the biological origins of immunoglobulins but also underscore the need to re-evaluate the roles of FLCs in health and disease.

The story of FLCs began in 1845 with the serendipitous discovery of Bence Jones proteins as a peculiar urinary protein that precipitated upon heating and redissolved upon boiling from a patient with multiple myeloma (MM) by the British physician Henry Bence Jones [2]. This finding marked the very first observation of FLCs, though their nature as immunoglobulin light chains remained elusive until decades later in the 20th century [3,4]. Comprising κ or λ subtypes, FLCs are synthesized in slight excess during immunoglobulin assembly, existing as monomers or dimers (λ-FLC mostly dimeric, κ-FLCs dimeric or monomeric) with short serum half-lives of 2–6 h, specifically of 2–4 h for κ-FLC and 3–6 h for λ-FLC [5,6,7]. Their rapid clearance involves glomerular filtration and proximal tubule reabsorption via the megalin–cubulin receptor complex, followed by lysosomal degradation [6].

Clinically, monoclonal FLCs have become important biomarkers of certain diseases, including hematopoietic malignancies (e.g., multiple myeloma, lymphocytic leukemia), amyloidosis, and light-chain deposition diseases [8,9,10,11], while polyclonal elevations correlate with autoimmune disorders (e.g., systemic lupus erythematosus, rheumatoid arthritis) and neurological diseases (e.g., multiple sclerosis, myasthenia gravis) [12,13,14,15,16]. In the FLC-related diseases mentioned above, monoclonal or polyclonal FLCs are considered to be produced by a single clone or multiple clones of B-lineage cells. However, the recent identification of non-B cell-derived FLCs has introduced a transformative dimension to this field. Unlike their B cell counterparts, these ectopically expressed FLCs exhibit distinct genomic architectures, physicochemical properties, and pathophysiological mechanisms, potentially contributing to specific pathologies such as promoting inflammation and carcinogenesis.

This review systematically examines the dual origins of FLCs, contrasting the gene organization, physicochemical properties, functional roles, and disease implications of B cell-derived versus non-B cell-derived FLCs. By elucidating their divergent molecular profiles and clinical impacts, we aim to provide a framework for understanding these molecules as dual-edged entities—both guardians of immunity and instigators of disease—while charting future directions for research and therapeutic innovation.

## 2. B Cell-Derived Free Light Chains

### 2.1. Genomic Organization and Rearrangement of B Cell-Derived FLCs

#### 2.1.1. Gene Structure of B Cell-Derived FLCs

Immunoglobulin light chains are composed of an N-terminal variable (V) region, which mediates antigen recognition through hypervariable sequences, and a C-terminal constant (C) region with conserved sequences within a species. Unlike heavy chain, light-chain V regions lack D segments and are encoded solely by V and J gene segments. During B cell development, V–J recombination assembles these segments, generating diverse light chains. This process is initiated by recombination signal sequences (RSS) flanking the V and J segments. Each RSS contains a conserved heptamer (5′-CACAGTG-3′) and nonamer (5′-ACAAAAACC-3′) motif separated by a 12- or 23-nucleotide spacer. The recombination-activating gene 1 (RAG1) and RAG2 recombinase induces double-strand breaks at these sites, enabling segment joining [17,18].

In humans, the immunoglobulin κ light-chain locus (IGK) is located at 2p11.2, and the λ light-chain locus (IGL) is located at 22q11.2. According to the IMGT databases (accessed 30 July 2025), the κ light-chain locus comprises *IGKV* genes grouped into seven subgroups (*IGKV1–IGKV7*, containing pseudogenes), followed by five functional *IGKJ* segments (*IGKJ1–IGKJ5*), and a single functional *IGKC* gene [19]. The λ light-chain locus comprises 11 subgroups of *IGLV* genes (*IGLV1–IGLV11*, containing pseudogenes), with seven downstream tandem *IGLJ-IGLC* cassettes, among which five are functional [19]. The genomic organization of *V*, *J*, and *C* gene segments in both light-chain loci, as well as the V–J recombination mechanism, is schematized in Figure 1.

#### 2.1.2. Biased Gene Segment Usage in Health and Disease

Under physiological conditions, the V–J arrangement and gene usage vary in individuals, while a limited set of immunoglobulin gene segments are actively expressed in healthy individuals. In contrast, pathological states exhibit marked selectivity, with distinct diseases showing preferential usage of specific gene segments. Details of V–J segment usage are shown in Table 1.

In healthy individuals, although there is no consensus on certain biased usage of gene segments, some common trends have been observed. Physiologically, predominant usage of segment *IGKV1*, *IGKV2*, *IGKV3*, and *IGKV4* in *IGKV* genes; *IGKJ1*, *IGKJ2*, and *IGKJ4* in *IGKJ* genes; *IGLV1*, *IGLV2*, and *IGLV3* in *IGLV* genes; and *IGLJ7* in *IGLJ* genes has been noted [20,21,22,23].

While under diverse pathological conditions, the preponderance of gene segment usage in each disease was reported. In patients with multiple myeloma (MM), *IGKV1*, *IGKV2*, and *IGKV3* are also widely observed for V–J arrangement. However, most studies do not indicate a preference for the usage of *IGKJ* or *IGLJ* genes [24,25,26,27]. In light-chain-derived (AL) amyloidosis, a preponderance of λ clones is observed, among which the three IGLV isotypes, including *IGLV3 (IGLV3-1)*, *IGLV6 (IGLV6-57)*, and *IGLV1 (IGLV1-44)*, are significantly over-represented, accounting for approximately 70% of the amyloidogenic light chains (LCs) [28,29,30,31]. Notably, the λ6 (*IGLV6-57*) germline is responsible for about 30% of all λ-AL cases [30]. In light-chain deposition disease (LCDD), usage of *IGKV4* (*IGKV4-1*), *IGKV1* (*IGKV1-5*), *IGKV3* (*IGKV3-11* and *IGKV3-15*), and *IGLV2* (*IGLV2-23* and *DPL12*) has been observed in the rearrangement of light chains [32,33,34,35]. In light-chain Fanconi syndrome (FS), FLCs of the IGKV1 subgroup are most frequently found [46,47,48]. Regarding autoimmune diseases, naïve B cells in patients with juvenile rheumatoid arthritis (JRA) show decreased usage of the downstream IGKV4-1 segment and almost exclusive usage of the most upstream V gene segments [36]. Additionally, in patients with JRA and systemic lupus (SLE), predominant usage of *IGKJ1* and *IGKJ2* is widely observed [36,37,38].

In conclusion, though normal FLC diversity arises under physiological conditions within a pool of favored gene segments, pathological conditions exhibit disease-specific recombination patterns.

### 2.2. Physicochemical Properties of B Cell-Derived FLCs

Despite substantial disparities and mutations in gene segment combinations leading to high variation in the amino acid sequence, three-dimensional structure, and resultant physicochemical properties, certain characteristics are observed under physiological and specific pathological conditions. Physiologically, the FLCs mainly exist as monomers or dimers, with most κ-FLCs being monomeric and most λ-FLCs being dimeric [49]. The V region of FLC consists of four framework regions forming a hydrophobic core, within which are three scattered complementarity-determining regions (CDRs) containing amphipathic amino acids such as tyrosine and tryptophan [50,51,52]. When an FLC dimer is formed, two identical light chains are packed end-to-end with substantially hydrophobic VL–VL and CL–CL interfaces, which can occur through interchain disulfide bonds or non-covalently via intermolecular electrostatic, hydrophobic, and hydrogen bonds [53]. The production of κ light chains is more prevalent than λ light chains, as reflected in a serum κ/λ ratio of 1.8:1. This results in the dominance of monomeric FLCs, as most κ-FLCs, are monomeric [54].

An increase in the proportion of dimeric and multimeric FLC is observed in various diseases, such as AL amyloidosis, MM, and multiple sclerosis [1,55,56,57]. Amyloidogenic FLCs are caused by the deposition of insoluble, non-branched β-sheet-rich fibrils initiated by the aggregation of misfolded FLCs. Amyloidogenic FLCs have been shown to exhibit less inherent thermodynamic stability, which is possibly correlated with the C region, and higher protein dynamics compared to their non-pathological counterparts FLCs [58,59,60]. Unlike AL, where fibril synthesis may be related to electrostatic interactions between light chains, hydrophobic interactions enhance amorphous precipitation in LCDD. The CDRs of FLCs in LCDD display distinct features, such as the replacement of hydrophilic amino acid residues with non-polar ones and an abundance of positive net charge [61]. Additionally, unusual FLC size and increased N-linked glycosylation have also been observed, which may be correlated with the propensity of monoclonal Ig LC to cause LCDD [35,62,63]. Generally, as a result of the physicochemical features above, B-FLCs may assemble into different physical forms, e.g., crystalline precipitates, non-amyloid granular deposits, organized amyloid fibrils, and obstructive distal tubular casts. B-FLCs existing in different physical forms are shown in Figure 2A.

### 2.3. Biological Functions and Pathogenesis Features of B Cell-Derived FLCs

FLCs transcend their traditional role as inert byproducts of immunoglobulin synthesis, demonstrating multifaceted biological functions, which position FLCs not merely as disease biomarkers but as bioactive molecules that directly contribute to pathology by driving inflammation via immune cell activation and complement, acting as chemoattractants, forming cytotoxic aggregates or amyloid fibrils, mimicking pathogenic autoantibodies, and causing direct cellular damage, particularly to renal tubules. This positions them as active participants in disease mechanisms across nephrology, hematology, and rheumatology. Details about the biological functions and pathogenesis features of B-FLCs are listed in Table 2 and illustrated in Figure 3A.

First of all, the antigen-binding site in intact immunoglobulin involves both heavy and light chains; the autonomous antigen-binding ability of FLCs remains controversial. Most studies show that isolated FLCs lack antigen specificity or exhibit significantly reduced affinity compared to intact antibodies [91,92,93]. Reconstitution experiments reveal that separated HCs regain antigen-binding capacity when paired with unrelated LCs, whereas isolated LCs fail to do so, suggesting HCs dominate antigen recognition [64,65,66]. On the other hand, a subset of studies report the affinity of antigen–FLC interactions [94,95,96]. Monomeric κ-FLCs were reported to bind with antigens, e.g., actin, myoglobin, TNP_25_-BSA, with different affinity [94]. A monoclonal antibody solely consisting of κ-FLCs was observed to recognize and bind to an unidentified molecule expressed by the nevus cells [95]. Furthermore, Schechter proved the retained antigen-recognition ability of FLCs isolated from complete antibodies [96]. The inconsistency in the results concerning the antigen-binding ability of FLCs may lie in the inherent discrepancies among immunoglobulins.

Furthermore, FLCs modulate innate immunity through interactions with key effector cells, including mast cells (MCs) and neutrophils. MCs are innate immune cells present in almost all tissues and play a crucial role in human allergic reactions through degranulation upon activation. The IgE-mediated activation pathway is the most well-known mechanism for mast cell activation [97]. However, FLCs bind MCs via an unidentified receptor (distinct from FcεRI/FcγRIII), enabling antigen-mediated crosslinking that triggers degranulation—a pathway independent of classical IgE activation [67,68]. Neutrophils are the first-line cells that respond and function in the innate immune defense system after bacterial infection. FLCs impair neutrophil chemotaxis and hexose uptake while prolonging survival by inhibiting apoptosis, contributing to chronic inflammation in conditions like uremia [69,70,71]. In chronic obstructive pulmonary disease, FLC–neutrophil binding induces CXCL8 production, amplifying inflammatory cascades [98].

Notably, pathogenic FLCs induce tissue damage, which occurs not only through physical damage or structural remodeling caused by extracellular deposition but also through direct toxicity. Among the various systems affected, the cardiac and renal systems are most commonly involved in light-chain deposition-related diseases, such as AL amyloidosis and LCDD. Cardiotoxic FLCs directly bind to several intracellular proteins, such as mitochondrial optic atrophy-1 (APO1), like protein and peroxisomal acyl-coenzyme A oxidase 1 (ACOX1), related to the survival and metabolism of mitochondrial proteins, generating excessive ROS that impair mitochondrial integrity [72,74]. Moreover, high-level cellular ROS induced by FLCs were observed to be involved in the FLC-induced p38 MAPK signaling cascade, eventually leading to cell apoptosis [75]. Additionally, FLCs disrupt autophagy by compromising lysosomal function, ultimately triggering cardiomyocyte death [76]. Pathogenic FLCs are involved in renal injury, but the pathophysiological mechanisms vary depending on the specific renal structure affected. In the glomerulus, fibrillar FLCs in AL amyloidosis and granular FLCs in LCDD deposit in the mesangial space, transform normal mesangial cells to a macrophage phenotype or a myofibroblastic phenotype, respectively, causing damage through altering the mesangial matrix, irregular mesangial proliferation and fibrosis, and apoptosis caused a reduction in mesangial cells in the late phage of disease [78,79,80]. In the tubular nephrons, proximal tubule-residing FLCs activate NF-κB, ASK1, and JAK-STAT pathways, promoting oxidative stress via hydrogen peroxide, apoptosis, and tubulointerstitial fibrosis [81,82,83]. In the tubules of distal nephrons, FLCs bind uromodulin to form obstructive casts, exacerbating tubular damage through mechanical blockage and rupture of the tubules and ensuing immune response [77].

## 3. Non-B Cell-Derived Free Light Chains

Emerging evidence challenges the classical paradigm that immunoglobulins are exclusively produced by B lymphocytes. Non-B cells, including epithelial cells, cardiomyocytes, and tumor cells, exhibit “ectopic expression” of free light chains (non-B-FLCs) with distinct genomic and functional properties.

### 3.1. Genomic Features of Non-B Cell-Derived FLCs

Under physiological conditions, gene rearrangement of light chains derived from certain non-B cells and tissues exhibits distinct characteristics. Huang et al. demonstrated that Igκ is expressed in primary spermatocytes and isolated epididymis epithelial cells of adult mice. Among the VκJκ sequences derived from primary spermatocytes, dominant rearrangement patterns include *Igkv*bv*9*/*Igkj1* and *Igkv21-10*/*Igkj2*, both of which exhibit >5% somatic hypermutation compared to germline sequences. In epididymal epithelial cells, the predominant rearrangement pattern was *Igkvbw20*/*Igkj5* [39]. Zhu et al. proved the conserved VκJκ usage in cardiomyocytes of adult mice, with *Igkv17-121*, *Igkv9-120*, and *Igkv14-100* being dominantly expressed [40]. Additionally, Igκ derived from primary hepatocytes of B cell-deficient μMT mice displayed distinct sequence characteristics, including preferential usage of *Igkv1-135*01*/*Igkj1* and *Igkv12–44*01*/*Igkj5* [41]. Furthermore, monocytes and neutrophils from patients with non-hematopoietic neoplasms frequently exhibited five rearrangement sets, including *IGKV3-20*01*/*IGKJ1*01*, *IGKV3-20*01*/*IGKJ3*01*, *IGKV1-27*01/IGKJ1*01*, *IGKV1-27*01*/*IGKJ4*01*, and *IGKV1-5*03*/*IGKJ3*01* [42].

In carcinomas, soft tissue tumors, and hematologic malignancies, non-B-FLCs also exhibit tumor-specific gene rearrangement and hypermutation patterns. Wang et al. observed a unique *IGKV4-1*/*IGKJ3* rearrangement pattern in Igκ light chains from multiple cancer cell lines, including HT-29 (colon cancer), A549 (lung cancer), HepG2 (hepatocellular carcinoma), HeLa MR (cervical cancer), and U2OS (osteosarcoma). These *IGKV4-1*/*IGKJ3*-FLCs show high homology to pathogenetic FLCs with identical rearrangements in LCDD and AL-AM [43]. Igκs derived from breast cancer, colon cancer, lung cancer, and hepatocellular carcinoma were further confirmed to predominantly utilize a unique *IGKV4-1*/*IGKJ3* rearrangement pattern [44]. Additionally, Igκ sequences from fibrosarcoma, leiomyoma, rhabdomyosarcoma, and leiomyosarcoma tissues were highly homologous to *IGKV3D-20*01*, with all Jκ sequences belonging to *IGKJ1* [45]. In acute myeloid leukemia (AML) myeloblasts, three recurrent *IGKV*/*IGKJ* rearrangements were identified, including *IGKV*1*5*03*/*IGKJ1*01*, *IGKV1-5*03*/*IGKJ3*01*, and *IGKV*1*-NL1*01*/*IGKJ5*01* [42]. Notably, the latter two show high inter-case homology.

Based on these findings, non-B cell-derived κ type FLCs exhibit cell-type-specific conserved rearrangement patterns under physiological conditions, accompanied by significant somatic hypermutation. Pathological FLCs display disease-characteristic rearrangements with *IGKV4-1*/*IGKJ3* in carcinomas and *IGKV*3*D-20*01* in soft tissue tumors, suggesting shared molecular mechanisms in a group of related pathological contexts. Details of V–J arrangement and gene segment usage are listed in Table 1.

### 3.2. Synthesis Regulation of Non-B Cell-Derived FLCs

Although mounting studies have validated the expression of Ig light chains in diverse non-B cell lineages, notably in almost all epithelial cancer cells, the genomic and molecular regulatory mechanisms underlying their production remain incompletely elucidated. Limited insights exist for light chains derived from nasopharyngeal carcinoma (NPC) cells and cardiomyocytes, with only few studies offering mechanistic clues.

A series of studies conducted by Cao et al. revealed the regulatory role of Epstein–Barr virus (EBV)-encoded latent membrane protein 1 (LMP1) on Ig light-chain expression. They identified functional regulatory elements, including the intronic enhancer (iEκ) and 3′ enhancer (3′Eκ), in specific NPC cell lines [99,100]. Mechanistically, LMP1 upregulates κ light-chain expression by enhancing 3′Eκ activity through ERK signaling-mediated activation of the transcription factor Ets-1 [100]. Concurrently, LMP1 promotes binding of heterodimeric NF-κB (p52/p65) and AP-1 (c-Jun/c-Fos) transcription factors to the iEκ enhancer region [101,102]. These findings challenge the classical view that Ig light-chain expression is strictly controlled by B cell lineage-restricted cis-regulatory elements, such as promoters/enhancers, thereby questioning the exclusivity of FLC production in B-lineage cells [99]. Interestingly, multiple immunoglobulins, such as Igκ-V8, Igκ-C, Igλ-V1, produced by cardiomyocytes were identified by Mehta‘s group. They demonstrated that light-chain expression is reduced in hearts of LOX-1 knockout mice, which is a cell surface lectin-like receptor for oxidized low-density lipoproteins (ox-LDL) critical in host defense. This loss was reversed upon exposure to angiotensin II-induced stress, implicating LOX-1 and hemodynamic stress in regulating cardiac Ig synthesis [103].

### 3.3. Physicochemical Properties and Morphological Forms of Non-B Cell-Derived FLCs

While the physicochemical properties of B cell-derived pathogenic FLCs in AL amyloidosis and LCDD are well characterized, non-B-FLCs remain understudied. Notably, Wang et al. reported that a free Igκ light chain with a unique Vκ4-1/Jκ3 rearrangement, derived from colon cancer, exhibits high homology to hydrophobic FLCs in AL amyloidosis and LCDD. These FLCs exist as monomers, dimers, and polymers, depositing both intracellularly and extracellularly in hydrophobic amorphous aggregates or fibrillar forms resembling extracellular matrix (ECM) proteins [43], suggesting their underlying pathological mechanisms and raising the question of whether these cancer-cell-deprived FLCs follow the similar regulation mechanism with their counterparts in AL amyloidosis and LCDD. In colitis and colitis-associated cancer, non-B-FLCs mainly exist as extracellular dimers [89]. Interestingly, in liver injury and hepatocellular carcinoma, dimeric and monomeric FLCs locate around the nucleus to form a point shape and display a filamentous network in hepatocytes and carcinoma cells [41,44]. Intracellular and extracellular pools of non-B-FLCs are depicted in Figure 2B.

### 3.4. Biological Functions and Pathogenesis Features of Non-B Cell-Derived FLCs

Non-B-FLCs exert diverse biological functions. Physiologically, non-B-FLCs are emerging as key players in tissue hemostasis, such as hepatocyte-derived Igκ in liver protection and lung epithelial Igκ in humoral immunity. Pathologically, non-B-FLCs drive tumor progression through multiple mechanisms, such as promoting inflammation and inflammation-mediated carcinogenesis, enhancing cancer cell survival, proliferation, invasion, and metastasis. Biological functions and pathogenesis features of non-B FLCs are listed in Table 2 and illustrated in Figure 3B.

#### 3.4.1. Physiological Functions of Non-B Cell-Derived FLCs

Yin et al. demonstrated that hepatocyte-derived Igκ protects against concanavalin A (ConA)-induced acute liver injury. In a ConA-induced liver injury model, mitochondrial apoptosis is triggered by cytochrome c release and caspase activation [104,105]. Notably, Igκ expression at both the protein and mRNA level increased in hepatocytes of ConA-challenged mice. Igκ knockout exacerbated liver injury via enhanced mitochondrial apoptosis, NF-κB inactivation, and JNK activation, suggesting Igκ’s protective role in hepatocyte survival and its potential contribution to liver hemostasis [41].

Similarly, Gao et al. identified that lung epithelial cells secrete immunoglobulins (IgM, IgA, Igκ, and IgG3) under serum-free conditions. Using an adeno-associated virus (AAV-Cre) to knock down Igκ specifically in Clara cells, they immunized mice with the T cell-dependent antigen NP-KLH. Igκ-deficient mice showed reduced NP-specific antibody production in lung epithelial cells, revealing Igκ’s essential role in long-distance immune responses and antibody secretion by non-B cells [88].

#### 3.4.2. Non-B Cell-Derived FLCs in Inflammation and Inflammation-Mediated Carcinogenesis

Nonallergic rhinitis (NAR) is an inflammatory disease lacking a clear allergic trigger from the history, with negative skin tests or negative specific IgE against common aeroallergens [106]. One possible explanation for this condition might be lie in non-IgE-mediated pathways involving FLCs. Redegeld et al. found that FLCs have a potential role in T cell-mediated immediate and delayed hypersensitivity immune responses, similar to IgE [87]. Significantly increased levels of FLCs in nasal secretions and enhanced FLC mRNA levels in nasal mucosa were discovered in NAR patients [107,108], which suggests that nasal mucosa cells may have the capacity to produce FLCs, although additional evidence is needed to confirm this possibility. Moreover, FLC expression showed concentrated localization in mast cells and a few distributions in eosinophils, indicating their possible role in local immune hypersensitivity and inflammation-mediated pathogenesis [107,108].

Chronic inflammation drives carcinogenesis through sustained tissue damage, oxidative stress, and dysregulated immune responses. Similarly, FLCs were found to promote murine colitis and colitis-associated cancer (CAC) carcinogenesis by inflammasome activation [89]. In both colitis and CAC mouse models, FLC was found to promote neutrophil activation. Moreover, active IL-1β and IL-18 levels are associated with increased FLC, along with more severe colitis damage and tumor formation. Conversely, using the FLC functional blocker peptide F991 significantly reduces activated IL-1β, IL-18, and activated caspase-1, while alleviating colitis progression and decreasing tumorigenesis [89]. Ig light-chain expression may serve as potential biomarkers of the stages in the process of inflammation-mediated carcinogenesis. Li et al. confirmed the Igκ light-chain constant region mRNA expressed by cervical epithelial cells and reported a significant increase in this mRNA level in dysplastic and carcinomatous cervical epithelial cells compared to cells with cervicitis, revealing a close association between Igκ constant region gene expression and cervical carcinogenesis development [109]. In addition, according to Kormelink et al., FLCs co-localized with mast cells in tumor-associated tissues from human breast, pancreas, lung, colon, skin, and kidney, but were absent in their healthy counterparts. Their study further revealed a positive correlation between FLC expression and aggressive tumor traits, such as increased proliferation, reduced apoptosis, and p53 mutation in basal-like breast cancers [110]. Moreover, research on a murine model of human malignant melanoma—which responds to anti-inflammatory therapy and is mast cell-dependent—suggests that FLCs promote tumor growth through FLC-activated mast cells, although specific mechanisms remain unclear [110,111,112,113].

Collectively, non-B-FLCs drive inflammation and tumor progression by activating inflammasome by enhancing the expression of cleaved caspase-1 and the maturation of IL-1β and IL-18. They also correlate with aggressive tumor traits and may serve as biomarkers or therapeutic targets to disrupt inflammation-to-cancer transitions.

#### 3.4.3. Pathological Role of Non-B Cell-Derived FLCs in Malignancies

FLCs derived from non-B cells exhibit tumor-promoting roles across diverse malignancies through distinct molecular mechanisms. In colon cancer, FLCs stabilize the anti-apoptotic protein Bcl-xL by directly interacting with it, thereby inhibiting mitochondrial apoptosis and enhancing tumor cell survival [114]. Recently, the free Igκ chain with a unique Vκ4-1/Jκ3 has been widely detected in multiple cancer cells, such as colon cancer, lung cancer, cervical cancer, proven to act as an ECM protein and an integrin β1 ligand in colon cancer [43]. These FLCs activate the FAK/Src signaling cascade, upregulate MMP-2 and MMP-9 expression, and drive cancer cell migration and invasion. Structural analysis reveals that the CDR2 region of Vκ4-1/Jκ3-FLCs is critical for integrin β1 binding, highlighting a unique structure–function relationship [43]. In hepatocellular carcinoma (HCC), hepatocyte-derived Igκ promotes tumor progression by modulating fatty acid metabolism. Igκ interacts with electron transfer flavoprotein A (ETFA), stabilizing its expression to enhance β-oxidation—a metabolic process critical for energy production in proliferating HCC cells. Depletion of Igκ in hepatocytes suppresses HCC growth in murine models, confirming its oncogenic role [44]. In addition, Wang et al. reported that the expression level of Igλ in cervical adenocarcinoma and cervical squamous cell carcinoma was higher than that in normal cervical tissue, and they obtained four potential tumor-derived Igλ-interacting proteins, RPL7, RPS3, and histones H1-5 and H1-6 [90]. RPL7 and RPS3, crucial for DNA damage repair, belong to the L30P and S3 ribosomal protein families, respectively. RPS3 also plays a key role in apoptosis, inflammation, tumorigenesis, and transcriptional regulation [115,116], interacting with p53 and lactate dehydrogenase to promote colon cancer cell proliferation, survival, migration, and invasion while reducing apoptosis [117]. Histone H1, a chromatin linker protein, includes 12 subtypes [118], among which H1-5 stabilizes higher-order chromatin structure and regulates gene expression, DNA repair, cell differentiation, proliferation, and metastasis [119,120]. While H1-6 facilitates chromatin decondensation for gene activation [121]. Tumor-derived Igλ interaction with these proteins may enhance oncogenic gene activation and metastasis by disrupting DNA repair via RPL7/RPS3 interference, inhibiting p53-mediated apoptosis, and promoting chromatin relaxation through H1-5/H1-6 interactions [90].

Beyond epithelial cancers, mesenchymal tumors such as sarcomas show a strong correlation between Igκ expression and proliferation markers, such as PCNA, Ki-67, cyclin D1, suggesting a pro-proliferative role for Igκ in these malignancies, though mechanistic insights remain limited [45]. In hematologic cancers like acute myeloid leukemia (AML), the *IGKV*1-5/*IGKJ*3*01 rearrangement in myeloblasts upregulates fLMP and CXCL12, chemotactic factors that drive AML cell migration and dissemination [42].

In conclusion, non-B-FLCs drive tumor progression across diverse cancers via distinct mechanisms and correlate with aggressive traits in mesenchymal tumors and hematologic malignancies. Especially, their structural specificity with Vκ4-1/Jκ3 rearrangement pattern and cross-tissue oncogenic roles highlight FLCs as multifunctional mediators and potential therapeutic targets in cancer biology.

## 4. Comparative Analysis of B Cell-Derived FLCs vs. Non-B Cell-Derived FLCs

Both B- or non-B-FLCs originate from V–J recombination of Ig light-chain genes consisting of V and J segments. However, their genomic preferences and structural features diverge under physiological or pathological conditions. B-FLCs exhibit disease-specific gene segment biases, such as the dominance of *IGLV6-57* (Vλ6) in AL [30]. However, non-B-FLCs lack consensus in gene segment usage among different lineages but frequently display a unique Vκ4-1/Jκ3 rearrangement pattern in various epithelial carcinomas, which is homologous to pathogenic B-FLCs in AL and LCDD [43]. Structurally, these Vκ4-1/Jκ3-FLCs can also form hydrophobic polymers that disrupt tissue architecture, similar to pathogenic B-FLCs.

Both B- and non-B-FLCs share similar roles in immune cell modulation, both involved in interacting with mast cells in diverse diseases, e.g., non-B-FLCs in tumor-associated inflammation, nonallergic rhinitis, and co-localization with mast cells in solid tumors (e.g., breast, lung), functionally echoing the role of B-FLCs in non-IgE-mediated mast cell activation [68,87,89,110]. However, significant divergence exists in the biological functions regarding other aspects. B-FLCs inhibit neutrophil apoptosis and exhibit enzymatic activities (e.g., anti-angiogenic, prothrombinase, and proteolytic functions), which have not been reported in non-B-FLCs [5,84,85,86]. Conversely, non-B-FLCs drive tumor progression via non-immunological pathways, such as stabilizing the anti-apoptotic protein Bcl-xL to enhance cancer cell survival or activating integrin β1/FAK/Src signaling to promote invasion and metastasis through MMP-mediated ECM remodeling [43,114].

Despite differences in disease associations, B- and non-B-FLCs share pathogenic mechanisms. Both induce extracellular deposition of hydrophobic aggregates, leading to structural damage like AL fibrils and tumor-associated ECM disruption [43,122]. In malignancies, the cellular origin of FLCs dictates their pathological impact. Non-B-FLCs in solid tumors (e.g., lung adenocarcinoma, colorectal cancer, and sarcomas) promote tumorigenesis by enhancing anti-apoptotic signaling via Bcl-xL stabilization, activating pro-invasive pathways via integrin β1/FAK/Src, rewiring metabolism via ETFA-dependent fatty acid oxidation, or disrupting DNA repair/epigenetic regulation via ribosomal/histone interactions and correlating with hyperproliferative phenotypes in mesenchymal tumors, etc. [43,44,45,89,90,110,114]. Conversely, B-FLCs in hematologic disorders serve as hallmarks of MM and Waldenström macroglobulinemia and directly cause organ damage through insoluble aggregate deposition, for example, tubular obstruction and tissue fibrosis, as well as molecular toxicity through activating NF-κB, complement pathways, and oxidative stress [49,122,123,124]. This dichotomy underscores the dual nature of FLC pathogenicity: non-B-FLCs drive epithelial/stromal tumor progression through oncogenic signaling, while B-FLCs mediate structural and inflammatory organ damage in hematologic malignancies.

## 5. Clinical and Therapeutic Perspectives

Currently, the clinical application of B-FLCs remains confined to their roles as a diagnostic, prognostic, and therapeutic marker in B cell clonal disorders, such as monoclonal gammopathy of undetermined significance (MGUS), MM, AL, and light-chain-related renal injuries. In these conditions, both the absolute levels of FLCs and the κ/λ ratio provide critical diagnostic and prognostic insights [125]. B-FLCs also show potential as biomarkers for autoimmune diseases (e.g., rheumatoid arthritis, primary Sjögren’s syndrome) and chronic inflammatory conditions, for example, chronic hepatitis B and diabetes, though further validation is required before clinical translation [126,127,128]. Emerging evidence suggests B-FLCs may serve as biomarkers for neurological disorders like myasthenia gravis and multiple sclerosis, but their routine use in these contexts is not yet established [13,14,15,16]. In contrast, non-B-FLCs remain poorly characterized pathologically, with no reported therapies targeting them to date.

Laboratory assessment of FLCs encompasses both qualitative and quantitative analyses of serum and urine samples. Qualitative or semi-quantitative methods primarily include protein electrophoresis and immunofixation electrophoresis [129]. For serum FLC quantification, the turbidimetry Freelite assay (The Binding Site), which employs with sheep polyclonal antibodies binding to hidden epitopes in the FLC constant region, was commercialized in 2001 [130]. Subsequent developments introduced alternative commercial serum FLC assays, including nephelometric methods like N-Latex FLC (Siemens; using monoclonal antibodies) and lateral flow methods like Seralite (Abingdon Health-Sebia; competitive inhibition lateral flow immunoassay), as well as the sandwich ELISA-based Sebia FLC assay (Sebia; using polyclonal antibody) [131,132]. Likely, urine FLC quantification is also possible using nephelometric and turbidimetric methods. However, the clinical utility of urinary FLC measurement remains limited. This is due to discordance between urinary FLC excretion patterns and serum FLC levels heavily influenced by renal function, combined with challenges in urine sample preparation and standardization [131]. Notably, mass spectrometry methods has recently demonstrated promising potential for the accurate and sensitive quantification of both serum and urine FLCs [133].

The focus and methodological considerations in FLC measurement vary significantly across diverse pathological contexts. In MM, the primary concerns are the absolute level of the involved FLCs reflecting selective κ or λ chain elevation due to plasma cell clonal expansion and the involved/uninvolved FLC ratio. Recommended laboratory testing includes serum and 24 h urine protein electrophoresis with immunofixation, alongside serum FLC quantification [131,132]. Crucially, when evaluating MM patients with chronic kidney disease (CKD), renal function must be factored into the interpretation of serum FLC levels, as these levels are influenced by renal clearance capacity. In multiple sclerosis (MS), the κFLC index calculated from cerebrospinal fluid (CSF) and serum FLC/albumin ratios is the validated diagnostic biomarker. Turbidimetric or nephelometric methods are recommended [133]. FLC measurement is not currently included in diagnostic guidelines, and specific methodological recommendations are lacking. However, it is worth noticing that for autoimmune diseases such as RA or SS, unlike the monoclonal FLCs observed in MM, the polyclonal FLC increase in autoimmune disorders necessitates assay compatibility with a broad range of light-chain epitopes.

With advancing research on both B- and non-B-FLCs, therapeutic strategies targeting these molecules hold significant promise. For B-FLC-associated diseases like MM and AL, current therapies focus on eliminating clonal plasma cells via alkylating agents, proteasome inhibitors, or anti-B cell antibodies, or reducing circulating FLCs via dialysis rather than directly targeting FLCs themselves, as their pathological effects largely stem from aggregation. However, recent breakthroughs include the anti-amyloid fibril antibody CAEL-101, which demonstrated organ response improvements in AL patients in Phase 1a/b trials, with ongoing Phase III studies, such as Mayo Stage IIIb [72,134,135]. Another promising avenue is blocking B-FLC interactions with mast cells to mitigate allergic and inflammatory responses [136]. Non-B-FLCs, however, represent a novel therapeutic frontier in oncology. These FLCs promote inflammation, carcinogenesis, and tumor progression in epithelial cancers. Among them, Vκ4-1/Jκ3-FLCs act as integrin β1 ligands, activating FAK/Src pathways and MMP-mediated ECM remodeling to drive metastasis; therefore, monoclonal antibodies targeting these FLCs show antitumor efficacy in preclinical models [43]. Moreover, Vκ4-1/Jκ3-FLCs interact with ETFA to enhance fatty acid β-oxidation, fueling tumor proliferation, and then disrupting the Igκ/ETFA axis suppresses HCC growth [44]. Targeting these interactions, such as integrin β1 or ETFA blockade, offers promising strategies for cancer treatment.

Therapies targeting B-FLCs and non-B-FLCs hold great translational promise for the future, yet several mechanistic questions need further exploration. For B-FLC-targeted therapies, challenges arise as the amino acid sequence and 3D structure of amyloid fibrils vary among patients, complicating the achievement of stable efficacy. Regarding non-B-FLCs, more research is needed to compare their structural properties and physicochemical characteristics with those of B-FLCs. Functionally, the exact interaction patterns and immune cell modulation mechanisms of B-FLCs and non-B-FLCs, particularly concerning mast cells and neutrophils, remain unclear. Moreover, the specific mechanisms by which FLCs promote cancer cell proliferation, metastasis, and immune escape need to be uncovered. For instance, in cervical cancers, how FLCs interact with proteins like RPL7, RPS3, H1-5, and H1-6, and the exact process of FLCs promoting sarcoma proliferation, are not yet fully understood [45,90]. In summary, while B-FLCs dominate hematologic disease management, non-B-FLCs represent untapped targets in solid tumors. Overcoming mechanistic and structural ambiguities will be pivotal to unlocking their full therapeutic potential.

## 6. Conclusions

B cell-derived and non-B cell-derived FLCs show both coherences and divergences in gene structures, physicochemical properties, biological functions, and pathogenic roles. They hold great clinical potential for accurate and sensitive diagnosis, targeted therapies, and timely monitoring of disease severity and prognosis across various diseases. However, research on FLCs is insufficient, particularly for non-B-FLCs, which have only garnered attention in recent decades. As exploration continues, a more in-depth understanding of FLCs and their translational potential is anticipated.

## Figures and Tables

**Figure 1 ijms-26-07607-f001:**
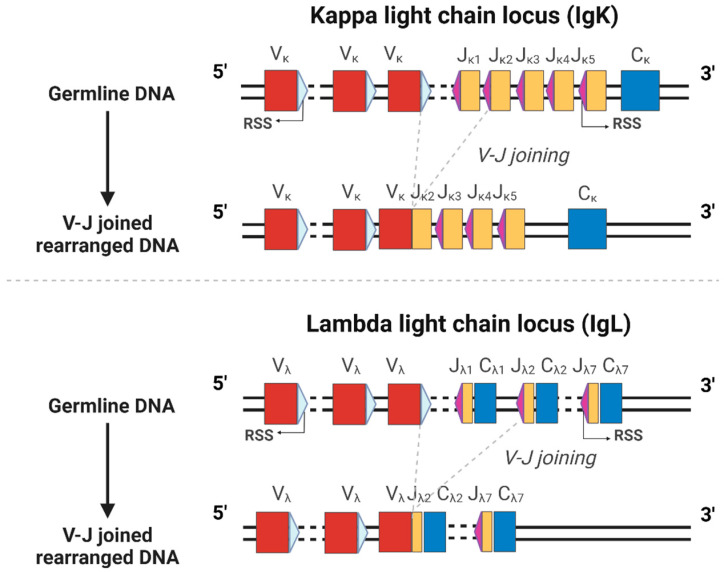
Schematic Diagram of κ-LC and λ-LC Locus Structures and V–J Recombination Mechanism. This diagram illustrates the genomic organization of variable (V), joining (J), and constant (C) gene segments in immunoglobulin light chains, along with recombination signal sequences (RSS) flanking V and J segments. The complex V gene clusters are depicted in simplified form. Within the IGK locus (κ-LC), the physical arrangements of 5 IGKJ segments and the single IGKC gene is shown. Within the IGL locus (λ-LC), the arrangement of the tandem array of IGLJ-IGLC units is shown. A simplified V–J recombination process, involving RSS (12-RSS flanking V segments and 23-RSS flanking J segments), is also schematized. Created in BioRender (BioRender.com). Li, L. (2025).

**Figure 2 ijms-26-07607-f002:**
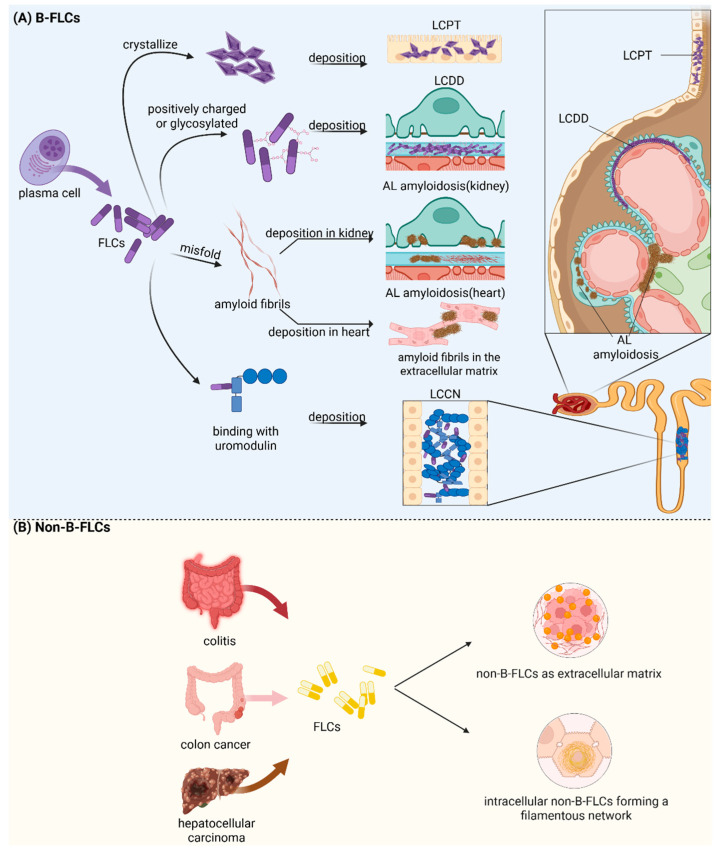
Morphology and Distribution of B Cell-derived and Non-B Cell-Derived Free Light Chains. This figure illustrates the typical morphological forms and tissue distributions of B cell-derived free light chains (B-FLCs) and non-B cell-derived free light chains (Non-B-FLCs). (**A**) B-FLCs predominantly deposit in the renal and cardiovascular systems, manifesting as crystalline inclusions in proximal tubule epithelial cells (light-chain proximal tubulopathy, LCPT), granular deposits in the glomerular basement membrane (light-chain deposition disease, LCDD), misfolded fibrils in the basement membrane, podocyte slit diaphragm, and cardiomyocyte extracellular matrix (AL amyloidosis), or uromodulin-bound casts in the distal tubule (light-chain cast nephropathy, LCCN). (**B**) Cancer-cell-derived or epithelial-cell-derived non-B-FLCs exist as extracellular amorphous monomers, dimers, polymers, and fibrils or are intracellularly located around the nucleus to display a filamentous network. Created in BioRender (BioRender.com). Li, L. (2025).

**Figure 3 ijms-26-07607-f003:**
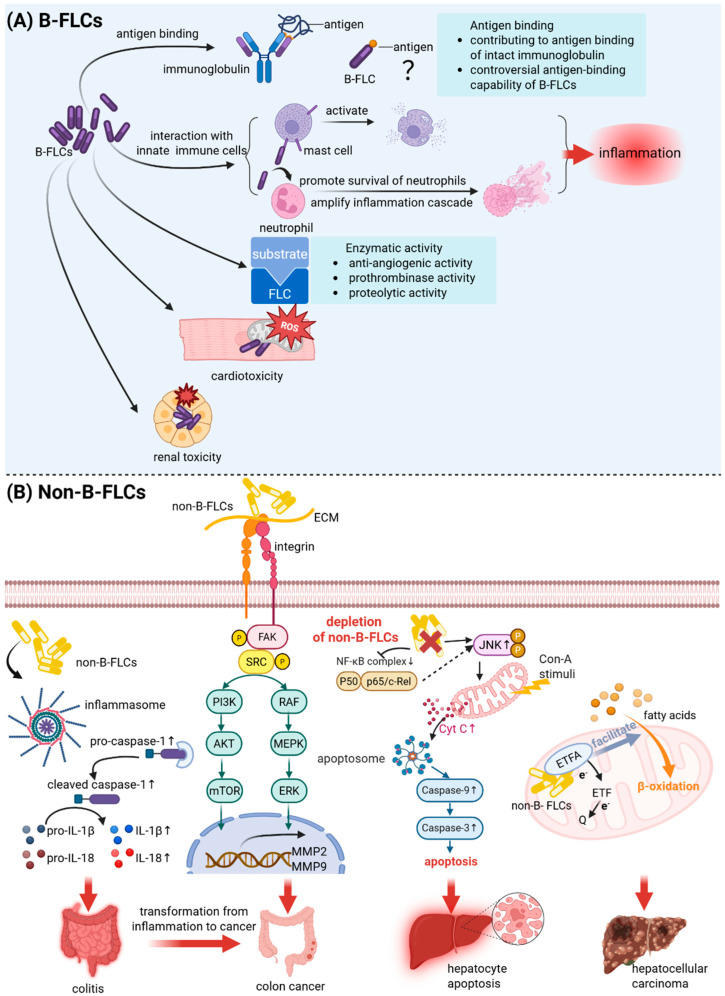
Diverse Biological Functions of B Cell-Derived and Non-B Cell-Derived Free Light Chains. This figure illustrates B-FLCs/Non-B-FLCs’ diverse biological functions with currently elucidated molecular mechanisms, both physiologically and pathologically. (**A**) Classical B-FLCs participate in the formation of the antigen-binding region of intact immunoglobulins, interact with innate immune cells, e.g., mast cells and neutrophils in inflammation, perform enzymatic activity, e.g., anti-angiogenic, prothrombinase, and proteolytic activity, as well as function in damage to the cardiovascular system and renal system. (**B**) Non-B-FLCs exhibit distinct mechanistic roles: inflammasome activation in colitis, integrin-mediated FAK/SRC pathway activation in colon cancer progression, hepatoprotection via apoptosis inhibition during liver injury and promotion of hepatocellular carcinoma through ETFA binding and stabilization to enhance β-oxidation. Created in BioRender (BioRender.com). Li, L. (2025).

**Table 1 ijms-26-07607-t001:** Biased Gene Segment Usage by B-FLCs and Non-B-FLCs in Health and Disease.

	FLC Origin	Condition	Gene Segment Usage	Refs.
B-FLCs	Plasma cells	Physiological	*IGKV1*, *IGKV2*, *IGKV3*, *IGKV4*;*IGKJ1*, *IGKJ2*, *IGKJ4*;*IGLV1*, *IGLV2*, *IGLV3*;*IGLJ7*	[20,21,22,23]
		Multiple Myeloma	*IGKV1*, *IGKV2*, *IGKV3*	[24,25,26,27]
		Systemic Amyloidosis	*IGLV3 (IGVL3-1)*, *IGLV6 (IGLV6-57)*, *IGLV1(IGLV1-44)*	[28,29,30,31]
		Light-Chain Deposition Disease	*IGKV4 (IGKV4-1)*, *IGKV1(IGKV1-5)*, *IGKV3(IGKV3-11* and *IGKV3-15)*, *IGLV2(IGLV2-23* and *DPL12)*	[32,33,34,35]
		Light-Chain Fanconi Syndrome	*IGKV1*	[32,33,34,35]
		Juvenile Rheumatoid Arthritis	*IGKJ1*, *IGKJ2*	[36,37,38]
		Systemic Lupus Erythematosus	*IGKJ1*, *IGKJ2*	[36,37,38]
Non-B-FLCs	Primary spermatocytes(mouse)	Physiological	*Igkvbv9/Igkj1*, *Igkv21-10/Igkj2*	[39]
	Epididymal epithelial cells(mouse)	Physiological	*Igkvbw20/Igkj5*	[39]
	Cardiomyocytes(mouse)	Physiological	*Igkv17-121*, *Igkv9-120*, *Igkv14-100*	[40]
	Hepatocytes(mouse)	Physiological	*Igkv1-135*01/Igkj1*, *Igkv12–44*01/Igkj5*	[41]
	Myeloblasts	Acute Myeloid Leukemia	*IGKV15*03/IGKJ1*01*, *IGKV1-5*03/IGKJ3*01*, *IGKV1-NL1*01/IGKJ5*01*	[42]
	Carcinoma cells	Lung Cancer, Hepatocellular Carcinoma, Cervical Cancer	*IGKV4-1/IGKJ3*	[43,44]
	Sarcoma cells	Fibrosarcoma, Leiomyoma, Rhabdomyosarcoma, Leiomyosarcoma	*IGKV3D-20*01*; *IGKJ1*	[45]

**Table 2 ijms-26-07607-t002:** Biological Functions of B-FLCs and Non-B-FLCs.

	FLC Origin	Biological Function	Refs.
B-FLC	Plasma cells	Contributing to the formation of the antigen-binding region	[64,65,66]
		Activating mast cells	[67,68]
		Promoting activation and inhibiting the apoptosis of neutrophils	[69,70,71]
		Cardiotoxicity	[10,72,73,74,75,76]
		Renal toxicity	[77,78,79,80,81,82,83]
		Enzymatic activity	[5,84,85,86]
Non-B-FLC	Nasal mucosal cells (not assured)	Potential interaction with mast cells	[87]
	Hepatocytes	Protecting hepatocytes	[41]
	Lung epithelial cells	Facilitating the secretion of TD-Ag-specific antibodies of lung epithelial cells	[88]
	Colon cancer cells	Promoting colon cancer	[43]
	Colon cancer cells	Promoting colitis-associated colon carcinogenesis	[89]
	Hepatocyte carcinoma cells	Promoting hepatocyte carcinoma	[44]
	Cervical cancer cells	Promoting cervical cancer	[90]
	Myeloblasts of AML	Promoting acute myelocytic leukemia	[42]
	Mesenchymal tumor cells	Correlating with hyperproliferative phenotypes in mesenchymal tumors	[45]

## Data Availability

The data generated in this study are available upon reasonable request from the corresponding authors.

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
