# Peer review of "B Cell-Derived and Non-B Cell-Derived Free Light Chains: From Generation to Biological and Pathophysiological Roles"

_ijms, 2025, doi:10.3390/ijms26157607_

Round 1
Reviewer 1 Report
Comments and Suggestions for Authors
This paper discusses the biological and pathological roles of free light chains (FLCs), which are traditionally known to be B cell products but are now recognized to also originate from non-B cells such as epithelial and carcinoma cells. The authors compare B cell-derived FLCs (B-FLCs) and non-B cell-derived FLCs (non-B-FLCs) in terms of their genetic features, structural properties, and functional roles in both health and disease. B-FLCs are involved in conditions such as multiple myeloma, AL amyloidosis, and autoimmune diseases, while non-B-FLCs have recently emerged as functionally relevant, particularly in cancer biology. Diagnostic and therapeutic implications are discussed.
This is a cutting-edge topic that expands the traditional immunological paradigm. The diagnostic potential of FLCs is promising, while therapeutic targeting remains uncertain. As this is a developing field, its future directions are yet to be determined. I believe the main purpose of this review is to raise awareness among immunologists and stimulate interest in further research.
The manuscript is well written, and I have no concerns.
Author Response
Comment: The manuscript is well written, and I have no concerns.
Response: We sincerely thank you for your positive evaluation and endorsement of our work. Your support highlights the clarity and scientific impact of our findings.
Reviewer 2 Report
Comments and Suggestions for Authors
I considered the manuscript entitled “B Cell-Derived and Non-B Cell-Derived Free Light Chains: From Generation to Biological and Pathophysiological Roles” by Linyang Li, et al, that is intended to be published in IJMS journal.
I very much enjoyed the manuscript. It is a well written review concerning B and non-B FLC structure, mechanisms, clinical and therapeutical significance. The manuscript is dense, well-organized and comprehensive.
To me, it only lacks an initial Figure where V and J zones are schematized. And, a Table describing the sequence of all the V and J gene sequences.
Author Response
Comment 1: It only lacks an initial Figure where V and J zones are schematized
Response 1:Thank you for this insightful suggestion, we have added Figure 1 (page 3) to the revised manuscript, illustrating the gene organisation of the κ-LC and λ-LC Loci and the V-J recombination mechanism.
Comment 2: a Table describing the sequence of all the V and J gene sequences
Response 2: We appreciate this valuable recommendation. While we acknowledge the importance of V/J gene sequences, comprehensive listings are available in the IMGT database. And full sequence compolation would exceed manuscript space limitations while adding minimal novel insight. To maintain conciseness and focus, we instead summarized key biased V-J recombination patterns in Table 1, and verified all gene loci against the latest IMGT database. We also updated descriptions in Section 1.1.1 (Gene Structure of B cell-Derived FLCs).
Reviewer 3 Report
Comments and Suggestions for Authors
The review is well organized and written.
I have only one minor comment regarding potential use of circulating FLCs assessment as biomarkers of disorders different from myeloma and MGUS into the clinical practice.
Are there any differences in biochemical assays among diverse FLCs according to their sources?
Author Response
Comment: Are there any differences in biochemical assays among diverse FLCs according to their sources?
Response: Thank you for raising this critical point, We have added two new paragraphs in Section 4 (Clinical and Therapeutic Perspectives) addressing methodological variations in FLC quantification (e.g., nephelometry vs. ELISA), source-dependent considerations (serum vs. urine; monoclonal vs. polyclonal FLCs), and clinical implications across diseases (MM vs. renal disorders vs. autoimmune conditions).
Round 2
Reviewer 2 Report
Comments and Suggestions for Authors
it is ok